# Possible magneto-mechanical and magneto-thermal mechanisms of ion channel activation in magnetogenetics

**Mladen Barbic***

Janelia Research Campus, Howard Hughes Medical Institute, Ashburn, United States

**Abstract** The palette of tools for perturbation of neural activity is continually expanding. On the forefront of this expansion is magnetogenetics, where ion channels are genetically engineered to be closely coupled to the iron-storage protein ferritin. Initial reports on magnetogenetics have sparked a vigorous debate on the plausibility of physical mechanisms of ion channel activation by means of external magnetic fields. The criticism leveled against magnetogenetics as being physically implausible is based on the specific assumptions about the magnetic spin configurations of iron in ferritin. I consider here a wider range of possible spin configurations of iron in ferritin and the consequences these might have in magnetogenetics. I propose several new magneto-mechanical and magneto-thermal mechanisms of ion channel activation that may clarify some of the mysteries that presently challenge our understanding of the reported biological experiments. Finally, I present some additional puzzles that will require further theoretical and experimental investigation.

DOI: https://doi.org/10.7554/eLife.45807.001

## Introduction

Interaction of biological systems with magnetic fields has puzzled and fascinated the scientific community for a long time (*Kirschvink and Gould, 1981*; *Kirschvink et al., 2001*; *Walker et al., 2002*; *Bazylinski and Frankel, 2004*; *Johnsen and Lohmann, 2005*; *Johnsen and Lohmann, 2008*; *Hore and Mouritsen, 2016*). While experimental evidence for magnetic sense in animals seems uncontroversial, the mystery of biophysical mechanism of its action remains unresolved. Despite the challenges in deciphering the fundamental operating principles of magnetic control of biological ion channels, cells, and organisms, the attraction of influencing biological systems with magnetic fields has remained strong. This is mainly due to the fact that external DC and AC magnetic fields easily penetrate biological tissue, are easily generated by current carrying wires or permanent magnets, and their properties and engineering design tools are well understood. These features of magnetic fields are commonly used in medical diagnostics applications such as Magnetic Resonance Imaging (MRI) (*McRobbie et al., 2017*), and there is a strong impetus to apply the same advantages of magnetic fields to control biological function, as is the case in Transcranial Magnetic Stimulation (TMS) (*Walsh and Cowey, 2000*).

Coupling modern genetic engineering techniques with the magnetic fields to influence biological activity has the potential to be a particularly robust way to combine the strength of both methods in control of cellular function. This is the approach of a recent technique development, commonly termed magnetogenetics, where thermo-sensitive and mechano-sensitive ion channels are genetically engineered to be closely coupled to the iron-storage protein ferritin (*Stanley et al., 2012*; *Stanley et al., 2015*; *Stanley et al., 2016*; *Wheeler et al., 2016*). Initial reports that introduced this new technology have received significant attention and commentary (*Anikeeva and Jasanoff, 2016*;

*****For correspondence:
barbicm@janelia.hhmi.org

**Competing interests:** The author declares that no competing interests exist.

*Nimpf and Keays, 2017*), as well as considerable criticism (*Meister, 2016*). The plausibility of presented mechanisms of mechanical and thermal activation of ion channels has been challenged on physical grounds (*Meister, 2016*), and the present state of reported experimental observations in magnetogenetics and basic magnetic physics arguments that challenge those observations remain in conflict.

The total energy of a spin system has terms related to the interaction of the individual spins with the external magnetic field, plus the interaction energy of the spins among themselves. The later contribution can be quite large, as occurs in ferromagnatism. This total energy must be compared with the thermal energy and, for interacting systems, the thermal energy may be too small to appreciably dephase the spins. Thus the possibility for experimenters to exploit the interaction of magnetic nanoparticles in vivo with external magnetic fields (~1T) is not unreasonable. Here, I present the case that the physical viability of magnetogenetics depends critically on many physical parameters of the basic control construct (iron-loaded ferritin protein coupled to the thermo-sensitive or mechano-sensitive ion channel in the cell membrane) that are presently not well known or understood, and therefore the physical possibility of magnetogenetics cannot yet be discounted and needs to be further explored. These critical parameters include the magnetism and magnetic spin configurations of iron atoms in the ferritin protein, as well as the realistic thermal, mechanical, and diamagnetic properties of ion channels and neural cell membranes coupled to the iron loaded ferritin. Additionally, I propose several new possible mechanisms of ion channel activation based on the magneto-caloric effect, mechanical cell membrane deformation by the diamagnetic force, and the mechano-thermal Einstein-de-Haas effect. I also discuss the fundamental magnetic moment fluctuations of the magnetic particle in ferritin and its presently unknown but potentially relevant effect on the ion channels in cell membranes. I emphasize that all of the arguments in this study are presented as theoretical exercises of what is in principle possible, both in terms of the magnetic materials synthesized within the ferritin protein and the experimentally applied settings. Further careful materials science and magnetometry studies need to be carried out to evaluate the viability of the described processes for in vivo magnetogenetics.

## Results

### Magnetism of iron in ferritin protein

The fundamental genetically engineered construct in magnetogenetics, as reported by the original articles (*Stanley et al., 2012*; *Stanley et al., 2015*; *Stanley et al., 2016*; *Wheeler et al., 2016*), is shown diagrammatically in *Figure 1*. Thermo-sensitive or mechano-sensitive ion channels of the TRP family of channels (TRPV1 and TRPV4) were genetically engineered to be closely coupled to the novel chimeric iron-loading protein ferritin. External DC or AC magnetic fields were applied to this construct in in vivo experimental settings, and it was preliminarily concluded that thermal or mechanical magnetic effects were likely responsible for the observed biological responses, as was intended by the genetic engineering methods. This interpretation has been challenged by *Meister (2016)* who has argued that the presently known magnetic force, torque, and heating mechanisms are many orders of magnitude smaller than necessary to activate ion channels and that the fundamental thermal energy level, $k_BT$, is much larger than the energy of any mechanism of magnetogenetics so far proposed. This criticism is fundamentally based on the assumptions on the number of iron atoms ($N = 2400$) in the ferritin protein and on the magnetic configuration in ferritin protein that assumes iron spins as a collection of independent non-interacting particles (paramagnetic configuration). Here, I explore what the physical consequences would be if those spin assumptions are expanded to include larger number of iron atoms in the ferritin protein (*Chasteen and Harrison, 1999*) ($N = 4500$) and spin configurations that allow for more strongly coupled iron spins within the ferritin core (ferromagnetic configuration).

It should first be noted that magnetic properties of iron are notoriously structure sensitive (*Coey, 2010*). Assigning the magnetic moment value to the iron atom between $0\mu_B$ to $5\mu_B$ (where $\mu_B$ is the Bohr magneton, $\mu_B = 9.27 \times 10^{-24}$ Am$^2$) and spin coupling configuration (paramagnetic, ferromagnetic, ferrimagnetic, antiferromagnetic, etc.) is highly variable and chemistry dependent (*Coey, 2010*). It should also be pointed out that the systematic and complete analysis of magnetism of the iron loaded ferritin core in all the reported magneto-genetics articles has not been performed,

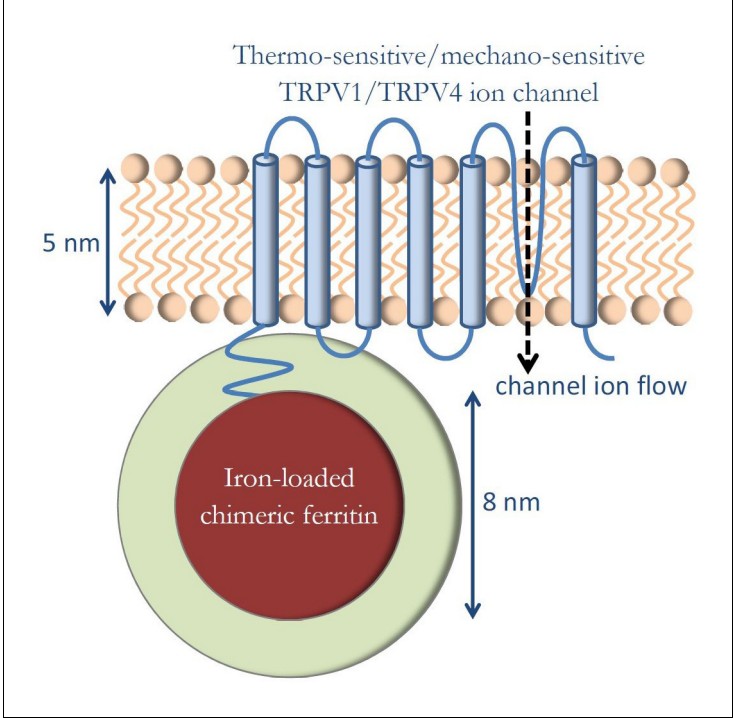

**Figure 1.** Genetically engineered construct in magnetogenetics. Thermo-sensitive or mechano-sensitive ion channel is closely coupled to the iron-loading protein ferritin. External DC or AC magnetic fields are applied in order to influence the ion channel function.
DOI: https://doi.org/10.7554/eLife.45807.002

and therefore the magnetism of the fundamental magneto- genetic ferritin construct is not presently known. It is generally believed that iron forms different mineral forms within ferritin (for example, magnetite $Fe_3O_4$ or maghemite $\gamma Fe_2O_3$). The original enhanced iron-loading chimeric ferritin, reported by *Iordanova et al. (2010)* was characterized only by NMR and MRI, and showed that one of the fused heavy H and light L subunits constructs (the L*H chimera) exhibited significantly enhanced iron loading ability. It is not clear from this report how many iron atoms are indeed loaded into the L*H chimera construct, what chemical structure iron forms in that construct, what the spin configuration of the iron atoms in that construct is, or if there is any shape or crystalline or any other magnetic anisotropy in the L*H ferritin (as the magnetic moment vs. magnetic field or magnetic moment vs. temperature measurements or chemical crystal analysis were not reported). Previous literature on such materials science, biomimetic synthesis, and magnetic investigations of iron loaded ferritin, both in physics and biology, is voluminous (*St. Pierre et al., 1986*; *St. Pierre et al., 1987*; *Awschalom et al., 1992a*; *Awschalom et al., 1992b*; *Gider et al., 1995*; *St. Pierre, 1996*; *Tejada et al., 1997*; *Wong et al., 1998*; *Quintana et al., 2004*; *Quintana et al., 2006*; *Collingwood et al., 2008*; *Quintana and Gutiérrez, 2010*; *Uchida et al., 2010*; *Plascencia-Villa et al., 2016*) and it is clearly imperative that such experimental techniques be applied to better understand the magnetism of the reported constructs in magnetogenetics studies.

In my analysis, I use the commonly stated maximum possible number of iron atoms in ferritin $N = 4500$ and assume atomic moment of $m_{Fe} = 5\mu_B$ per iron atom (the highest value reported for iron in oxide form, Table 3.5 in *Coey, 2010*), with the understanding that the actual number of iron atoms in ferritin in all the magnetogenetics reports is not presently known.

*Figure 2* diagrammatically shows the three distinct spin coupling configurations of iron atoms in ferritin that I consider as reasonable possibilities. *Figure 2a* represents N irons spins in a conventional paramagnetic ferritin state (*Meister, 2016*) where all the spins are magnetically independent from one another and non-interacting. *Figure 2c* shows the case of ferromagnetic coupling between N iron spins where all the spins are strongly coupled by the exchange interaction and magnetically

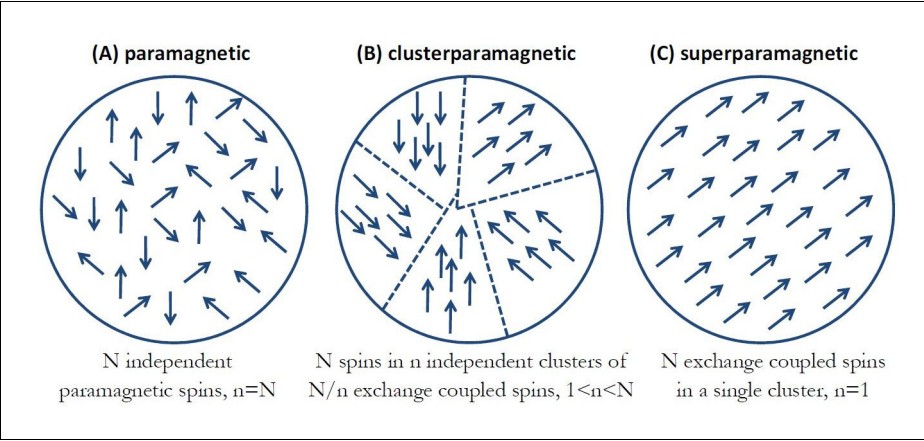

**(A) paramagnetic**

**(B) clusterparamagnetic**

**(C) superparamagnetic**

N independent
paramagnetic spins, n=N

N spins in n independent clusters of
N/n exchange coupled spins, 1<n<N

N exchange coupled spins
in a single cluster, n=1

**Figure 2.** Three distinct spin coupling configurations of iron atoms in ferritin. (a) Paramagnetic state where N irons spins are magnetically independent from one another and non-interacting. (b) Clusterparamagnetic state where N iron spins are separated into n independent clusters of N/n exchange coupled spins. (c) Superparamagnetic state where all N spins are strongly coupled by the magnetic exchange interaction and behave as a single macro-spin.
DOI: https://doi.org/10.7554/eLife.45807.003

behave as a single macro-spin in what is commonly termed a superparamagnetic state of the magnetic particle (*Coey, 2010*). In *Figure 2b*, I consider a state that lays in between these two extremes. In this configuration, the N iron spins are separated into n independent clusters of N/n exchange coupled spins, and I will call this spin configuration a clusterparamagnetic state. There have been numerous experimental investigations on ferritin, both structural and magnetic, that indicate that such spin structure might be the most probable one in some cases (*Haggis, 1965*; *St. Pierre et al., 2001*; *Brem et al., 2006*; *Gálvez et al., 2008*; *Martínez-Pérez et al., 2010*; *López-Castro et al., 2012*). For simplicity of analysis, I will assume that all of the n magnetic clusters in this spin configuration of the particle have the same number N/n of iron spins, and that the magnetic moments of the n clusters within the ferritin particle are magnetically independent (non-interacting). It should be clear that *Figure 2b* represents the most general spin configuration with n clusters of N/n spins in each cluster, while the paramagnetic state of *Figure 2a* is a special case with n = N and the superparamagnetic state of *Figure 2c* is a special case where n = 1. I will assume the physiological temperature of T = 37°C = 310K throughout. The thermal average magnetic moment in external magnetic field B of a single cluster of N/n iron spins in the ferritin particle is classically described by the Langevin function (*Coey, 2010*):

$$m_{cluster} = \frac{N}{n} \cdot m_{Fe} \cdot \left[ coth(x) - \frac{1}{x} \right] \qquad (1)$$

where parameter $x = \frac{\frac{N}{n} \cdot m_{Fe} \cdot B}{k_B \cdot T}$, and the total thermal average magnetic moment $m_{TOT}$ of the ferritin particle along the field direction is:

$$m_{TOT} = n \cdot m_{cluster} \qquad (2)$$

*Figure 3a* shows the resulting ferritin particle magnetic moment vs. magnetic field at T = 310K as a function of different cluster numbers and for the experimentally attainable laboratory magnetic fields (0 to 2 Tesla) generated by either electro-magnets, superconducting magnets, or permanent magnets.

It is immediately apparent that the assumption about the spin configurations of iron atoms in the ferritin protein has dramatic effects on the total magnetism of the particle. For the paramagnetic (*Figure 2a*) spin arrangement of the particle (n = N = 4500, light blue curve in *Figure 3a*), the magnetic susceptibility is low, and as an example, the total magnetic moment of the particle at a representative field of 1 Tesla is m = 2.4×10$^{-22}$ (Am$^2$), consistent with the previous paramagnetic ferritin assumption (*Meister, 2016*). However, for the superparamagnetic (*Figure 2c*) spin arrangement

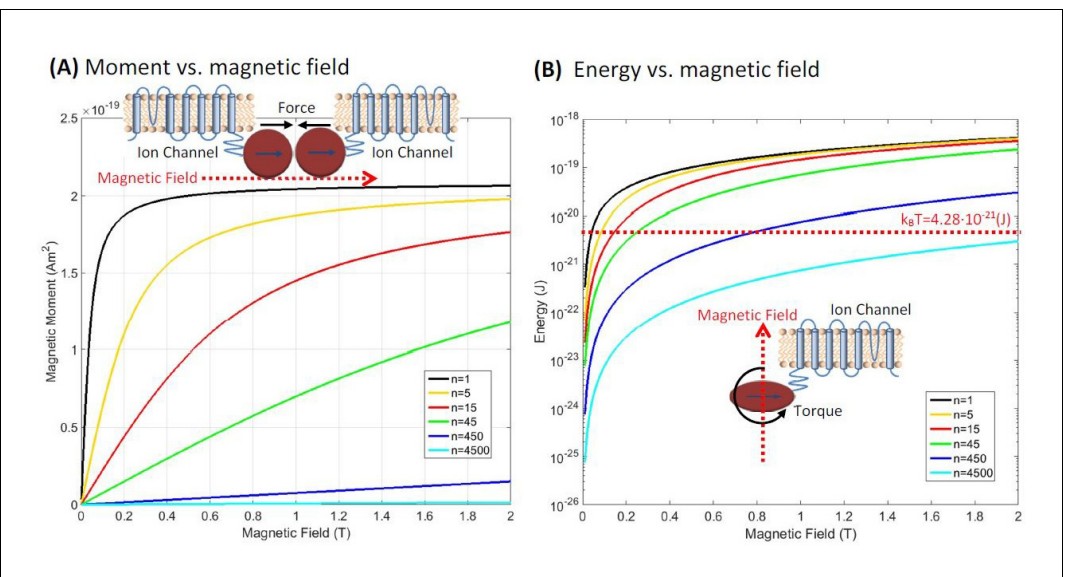

**Figure 3.** Magnetic properties of clusterparamagnetic ferritin. (a) Ferritin particle magnetic moment vs. magnetic field as a function of different cluster numbers n of N/n exchange coupled spins in a particle of N spins. For the superparamagnetic spin arrangement of Figure 2c (n=1, black curve), the particle moment saturates at relatively low magnetic fields and the magnetic moment is three orders of magnitude higher than for the paramagnetic state of Figure 2a (n = 4500, light blue curve). The curves for the clusterparamagnetic configurations of Figure 2b (n = 5, 15, 45, 450) are also shown. The inset in (a) shows the attractive force configuration between two ferritins. (b) The interaction energy magnitude (E = m·B) of the iron loaded ferritin as a function of the external magnetic field. For modest clustering of iron spins into n clusters of N/n exchange coupled spins the interaction energy is above $k_BT$ in moderate magnetic fields. The maximum theoretically possible torque on an anisotropic ferritin particle $\vec{\Gamma} = \vec{m}X\vec{B}$, shown diagrammatically in the inset of (b), has interaction energy above $k_BT$.

DOI: https://doi.org/10.7554/eLife.45807.004

(n = 1, black curve of *Figure 3a*), the particle moment saturates at relatively low magnetic fields, well below 1 Tesla, where the magnetic moment has the value of m = $2\times10^{-19}$ (Am$^2$), three orders of magnitude higher than for the paramagnetic state. This is the classic superparamagnetic particle behavior (*Bean and Jacobs, 1956*) where N = 4500 spins are uniformly exchange coupled and act as a giant single paramagnetic spin. The other curves in *Figure 3a* show the magnetic moment for the clusterparamagnetic (*Figure 2b*) spin configurations (with cluster number n = 5, 15, 45, 450) indicating that even modest clustering of N = 4500 spins into n clusters of exchange coupled N/n spins can significantly increase the magnetic moment of the ferritin particle in reasonable laboratory magnetic fields.

Most experimental reports on ferritin magnetism indicate paramagnetic particles (*Meister, 2016*). However, that is not necessarily known for the ferritin construct reported by *Iordanova et al. (2010)* (and all the subsequent magnetogenetics reports) for which the magnetization curves of the particles are not reported. In fact, there have been experimental reports on iron-loaded ferritin where the magnetization measurement closely follows the superparamagnetic curve (black line in *Figure 3a*) (*Bulte et al., 1994*; *Moskowitz et al., 1997*). In the present analysis, the saturation magnetic moment of a superparamagnetic particle of N = 4500 iron atoms with moments of 5$\mu_B$ per atom is $m_{TOT}$ = 22500 $\mu_B$ = 2.1 x $10^{-19}$ (Am$^2$), consistent with the numbers reported in experiments by *Bulte et al. (1994)* and *Moskowitz et al. (1997)*, suggesting that the superparamagnetic state of iron-loaded ferritin might be possible. It is also instructive to consider the saturation magnetization, $M_S$, of magnetite in my calculation by dividing the particle saturation moment $m_{TOT}$ by the volume V of the 8 nm diameter particle ($M_S = m_{TOT}/V$) which results in $M_S$ = $7.8\times10^5$ (A/m). Commonly reported bulk magnetite magnetization is slightly lower at $M_S$ = $4.8\times10^5$ (A/m) (*Coey, 2010*), but it should be noted that several reports (*Arora et al., 2008*; *Orna J et al., 2010*; *Guan et al., 2016*)

indicate that magnetite magnetization increases significantly in the size range below 10 nm and reaches the value of $M_S = 1\times10^6$ (A/m). Therefore, $M_S = 7.8\times10^5$ (A/m) value I use for 8 nm diameter ferritin particle of N = 4500 iron spins with $5\mu_B$ per iron atom appears reasonable.

## Force and torque on iron-loaded ferritin

I now reconsider the possible forces in magnetogenetics on a ferritin particle (and therefore on the mechano-sensitive ion channel). In the most optimistic case where the particle is at the entrance to the bore of a superconducting MRI magnet, where the magnetic field gradient could be on the order of $\nabla B = 50$ (T/m). The force on the paramagnetic ferritin would be on the order of $F = m \cdot \nabla B = 10^{-20}$ (N), while the force on the superparamagnetic ferritin would be on the order of $F = 10^{-17}$ (N). The minimum required force to mechanically activate ion channels (*Milo and Phillips, 2016*) is on the order of 1 (pN) = $10^{-12}$ (N). This confirms that the force from an externally applied magnetic field gradient on iron loaded ferritin, even the superparamagnetic one, is too low to be effective (*Meister, 2016*).

The situation is quite different when one considers the mutual attractive force between two iron loaded ferritin particles that are both in a superparamagnetic spin configuration, as depicted in the inset of *Figure 3a*. In an external magnetic field of 1 Tesla, both ferritin particles will be magnetically saturated, as *Figure 3a* shows. For 8 nm diameter (radius r = 4 nm) ferritin particle, the maximum magnetic field gradient on the surface of the particle is:

$$\nabla B = 2 \cdot \mu_0 \cdot M_S / r \quad \text{where} \mu_0 = 4\pi \cdot 10^{-7} (\text{Tm/A}) \tag{3}$$

and has the value of $\nabla B = 4.9\times10^8$ (T/m), many orders of magnitude larger than anything achievable with external laboratory magnetic field gradients. Such a large magnetic field gradient from one ferritin particle acting on the second ferritin particle (and therefore on the mechanosensitive ion channels coupled to the ferritin particles) results in the maximum possible force on the order of $F = 10^{-10}$ (N) = 100 (pN), well above the required level for activating an ion channel. Therefore, if the construct reported by *Iordanova et al. (2010)* and used by the magnetogenetics reports is similar in its magnetism or even more enhanced than what was reported by other superparamagnetic ferritin results of *Bulte et al. (1994)* and *Moskowitz et al. (1997)*, then there is at least the theoretical plausibility that the two ferritin particles pulling on each other could result in a sufficient force to activate mechano-sensitive ion channels to which the ferritin particles are coupled.

*Figure 3b* shows the interaction energy magnitude (E=m·B) of the iron loaded ferritin configurations of *Figure 2* in the external magnetic fields between 0 and 2 Tesla. On the same semi-log plot I indicate the thermal energy level at the physiological temperature of T = 310K, $k_BT = 4.28\times10^{-21}$ (J). As has been pointed out for the paramagnetic ferritin particle (*Meister, 2016*) (n = 4500 light blue curve in the plot), that interaction energy is lower than $k_BT$. However, it is interesting that even for modest clustering of N = 4500 iron spins into say n = 450 clusters of N/n = 10 exchange coupled spins each (n = 450 dark blue curve in the plot) the interaction energy of the particle rises above $k_BT$ level in a field of 1 Tesla. The magnetic energy diagram of Figure 3b is also interesting in that the maximum theoretically possible torque on an anisotropic ferritin particle (*Meister, 2016*) can easily be evaluated since torque $\vec{\Gamma} = \vec{m} X \vec{B}$. Therefore, there is in principle a theoretical possibility that an external magnetic field can exert a sufficient torque on an anisotropic ferritin particle (on the order of $10^{-19}$ N·m in a 1 Tesla external field) to activate mechano-sensitive ion channels.

## Diamagnetic force on ion channel from iron-loaded ferritin

I now consider the force due to the magnetic fields and field gradients from the ferritin particle itself on the intrinsically diamagnetic mechano-sensitive ion channel and neural cell membrane. I suggest that this diamagnetic repulsive force might be sufficient to mechanically deform the ion channel and affect its function, as I diagrammatically describe in *Figure 4*. Diamagnetism is generally considered the weakest form of known magnetism (*Coey, 2010*; *Yamaguchi and Tanimoto, 2006*), but it can have surprising and dramatic mechanical effects on biological materials (most of which are diamagnetic), such as levitation (*Geim et al., 1999*; *Simon and Geim, 2000*) or restriction of water flow (*Ueno and Iwasaka, 1994a*; *Ueno and Iwasaka, 1994b*), if the conditions of large magnetic fields and field gradients are simultaneously present. The diamagnetic force per unit volume on a diamagnetic material is (*Simon and Geim, 2000*):

$$F/V = -\frac{|\chi|}{\mu_0} B(r)\nabla B(r) \tag{4}$$

It is apparent that the critical parameter for generating appreciable force on a material with diamagnetic susceptibility $\chi$ is the product of the magnetic field and the magnetic field gradient $B(r)\nabla B(r)$. In laboratory settings at the edge of a superconducting magnet bore this parameter can have values on the order of $10^3$ (T²/m); (**Simon and Geim, 2000**).

I calculate the value of this parameter at the location of the ion channel/cell membrane that is closely coupled to the superparamagnetic ferritin particle, as shown in **Figure 4a**. In an externally applied uniform magnetic field of $B_{ext}$ = 1 Tesla the superparamagnetic particle magnetic moment will be saturated (as shown in **Figure 3a**), and the ion channel in the cell membrane will experience the total combined magnetic field from the saturated magnetite particle $B_{particle}$ and the external magnetic field $B_{ext}$ as shown in **Figure 4a**. The magnetic field on top of the saturated 8 nm diameter magnetite particle in my model is:

$$B_{particle} = \frac{2}{3}\cdot\mu_0\cdot M_S = 0.65 \ (T) \tag{5}$$

and the total maximum field seen by the ion channel and the cell membrane is $B = B_{ext}+B_{particle}$ = 1.65 (T). The ion channel in the cell membrane will also be under the influence of the particle gradient magnetic field of $\nabla B = 4.9\times10^8$ (T/m) (from **Equation (3)**). This results in the critical parameter in the diamagnetic force calculations of $B\nabla B = 8.1\times10^8$ (T²/m), many orders of magnitude larger than anything available in the common laboratory settings (**Simon and Geim, 2000**). This combination of magnetic fields and magnetic field gradients from the ferritin particle will generate a repulsive diamagnetic vector force on the ion channel/cell membrane, as shown diagrammatically with red arrows in Figure 4b. It is particularly intriguing that the diamagnetic repulsive force from the ferritin particle is exerted on the mechanosensitive ion channel in a neural cell membrane that is known to be mechanically extremely soft (**Tyler, 2012**). The consensus that is emerging from the studies of mechanical properties of ion channels and cell membranes (**Lu et al., 2006**; **Sánchez et al., 2008**; **Wu et al., 2018**) is that neural cells have extremely low Young's modulus, E, on the order of E = 100 (Pa), the lowest of any known materials. The final parameter in the diamagnetic force **Equation (4)** is the diamagnetic susceptibility $\chi$ of the ion channel and the cell membrane. It is reasonable to assume that this number is similar to that of water (**Coey, 2010**), on the

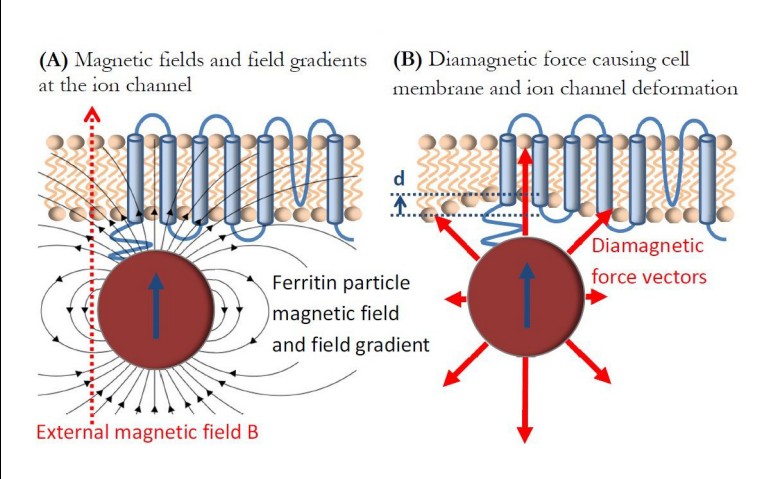

**Figure 4.** Diamagnetic force deformation of ion channel and cell membrane. (a) Diamagnetic ion channel in the cell membrane experiences magnetic fields from the ferritin particle and the externally applied magnetic field B, as well as the large magnetic field gradient from the ferritin particle. This results in the repulsive diamagnetic force on the ion channel and the cell membrane in (b) that is sufficient to potentially mechanically deform them and affect the ion channel function.

DOI: https://doi.org/10.7554/eLife.45807.005

order of $\chi$ = -1x10$^{-5}$, but one should not dismiss the possibility that ordered lipid cell membrane domains could have an order of magnitude higher value of diamagnetic susceptibility (*Braganza et al., 1984*; *Ginzburg et al., 1984*).

Estimating the resulting deformation of the ion channel and cell membrane, as shown in *Figure 4b*, due to the diamagnetic force of *Equation (4)* is extremely difficult, as both the magnetic fields and the magnetic field gradients from the magnetite particle are spatially rapidly varying in both direction and magnitude, and cell membrane and ion channel nanomechanics and diamagnetism are poorly understood. However, using contact mechanics equations (*Sánchez et al., 2008*) one can give an order of magnitude estimate for the cell membrane deformation d in *Figure 4b* using the listed magnetic, mechanical, and diamagnetic parameters for the ferritin construct and the cell membrane. The vertical displacement d due to a force F over an area A = $\pi r^2$ over a circular region of radius r = 4 nm on an elastic flat half space of Young' Modulus E and Poisson ratio $\nu$ (assumed to be $\nu$ = 0.5 for the neural cell [*Tyler, 2012*]) is estimated to be (*Sánchez et al., 2008*):

$$d = \frac{2(1-\nu^2)r}{E}F/A \qquad (6)$$

Combining *Equation (5)* with *Equation (6)*, using the listed parameters, and assuming for simplicity that the diamagnetic force (of *Equation (5)*) is applied uniformly only across 1 nm of cell membrane thickness (due to the sharp drop of fields and field gradients from the magnetite particle), results in the deformation distance d = 0.4 nm = 4 Å. This is of course only a crude estimation, but such an Angstrom-scale deformation represents a significant fractional change of the overall cell membrane thickness, and is known to be sufficient to affect ion channel behavior (*Tyler, 2012*; *Hamill and Martinac, 2001*; *Reeves et al., 2008*) and is also quite likely to do so in mechano-sensitive magnetogenetics constructs.

## Magneto-caloric effect in iron-loaded ferritin

Heating of iron loaded ferritin by AC magnetic fields has also been considered as one of the mechanisms for thermo-sensitive ion channel activation and criticized to also be implausible (*Meister, 2016*). I now consider a thermal mechanism that to my knowledge has not been considered for the clusterparamagnetic spin arrangement in magnetogenetics: the DC magneto-caloric effect in the iron-loaded ferritin particle. Magneto-caloric effect refers to the heating and cooling of magnetic materials by DC magnetic fields (*Coey, 2010*). It is fundamentally based on the physical principle that an ensemble of magnetic spins in zero magnetic field is fluctuating and therefore maximally randomized and in a high entropy state. Upon application of a polarizing magnetic field the magnetic spins align in the field which lowers the entropy of the spin ensemble. In an adiabatic process where there is no exchange of heat with the environment, this change in spin ensemble entropy has to be compensated for by the exchange of energy between the spin ensemble and the magnetite particle lattice, resulting in the change of temperature of the particle. This process of magnetic adiabatic cooling has been used in low-temperature physics for a long time (*Giauque, 1927*) to obtain milliKelvin temperatures in paramagnetic salt powders and sub-milliKelvin temperatures with nuclear spins in metals (*Simon, 1952*). Although such magnetic adiabatic energy transfers are generally performed only at cryogenic temperatures, *McMichael et al. (1992)* have pointed out that this process can be potentially performed at higher temperatures if the spins are segregated and coupled into superparamagnetic clusters (such as shown in the spin configuration of *Figure 2b*). For a given magnetic field B and temperature T, there is an optimal clusterparamagnetic size that will generate maximum entropy change and energy transfer from the spin ensemble to the magnetite particle lattice (*McMichael et al., 1992*). This hypothesis seems to have been experimentally confirmed (*McMichael et al., 1993*; *Shao et al., 1996*). Here I investigate the magneto-caloric energy transfer values for the specific case of the clusterparamagnetic iron loaded ferritin (*Figure 2*) with N = 4500 iron atoms with 5$\mu_B$ atomic moments at a physiological temperature of T = 310K.

*Figure 5* diagrammatically shows the basic process. In zero magnetic field, n clusterparamagnetic moments (configuration of *Figure 2b* with N/n spins in each cluster) of the particle are randomly fluctuating (*Figure 5a*) and have a high magnetic entropy state. Upon application of an external magnetic field on the order of 1T, the clusterparamagnetic moments will mostly align with

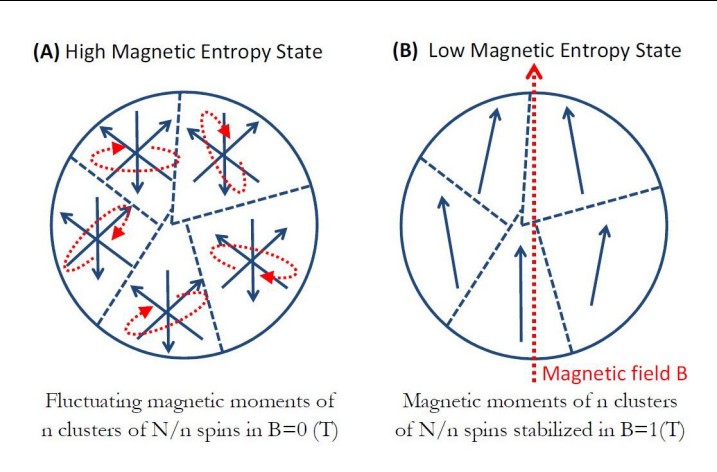

**Figure 5.** Magneto-caloric effect in clusterparamagnetic ferritin. (a) In zero magnetic field, n clusterparamagnetic moments of N/n exchange coupled spins are randomly fluctuating and have high magnetic entropy. (b) Upon application of external magnetic field B the clusterparamagnetic moments align with the field and have low entropy. In the adiabatic process this change in spin entropy is compensated for by the exchange of energy between the spin ensemble and the magnetite particle lattice. For a given magnetic field B and temperature T, there is an optimal clusterparamagnet size that will generate maximum entropy change and energy transfer to the magnetite particle lattice.
DOI: https://doi.org/10.7554/eLife.45807.006

the magnetic field (*Figure 5b*) and go into a low entropy state. The classical entropy change in such a process for n-cluster particle (with N/n spins in each cluster) is (*McMichael et al., 1992*):

$$\Delta S = n \cdot k_B \cdot \left[ 1 - x \cdot coth(x) + ln\left( \frac{sinh(x)}{x} \right) \right] \qquad (7)$$

where, again, parameter $x = \frac{\frac{N}{n} \cdot m_{Fe} \cdot B}{k_B \cdot T}$. This magnetic entropy change results in the energy transfer to the magnetite particle lattice of $\Delta E = T \cdot \Delta S$.

*Figure 6* shows the numerical calculation results for that magneto-caloric energy change as a function of the number of clusters n in the particle. For magnetic field change of 1 Tesla (black points in *Figure 6*), the ferritin particle with N = 4500 iron spins that separate into n = 15 equal clusters of N/n = 300 exchange coupled spins in each cluster will provide the maximum energy transfer to the magnetite particle lattice. In the same plot I indicate the thermal energy level $k_B T$ that reveals the interesting features in this analysis. For the paramagnetic spin state of the ferritin particle (N = n = 4500 point in *Figure 6*) no value of magnetic field is sufficient to achieve magneto-caloric particle energy change near the level of $k_B T$. However, if N = 4500 iron spins conceivably cluster into n = 100 (or fewer) clusters of N/n exchange coupled spins each, the magneto-caloric energy transfer from the spin ensemble to the magnetite particle lattice is higher than the $k_B T$ level, including for the superparamagnetic state (n = 1 point in *Figure 6*). It is not clear how this magneto-caloric energy transfer from the clusterparamagnetic spin ensemble of the magnetite particle to the thermo-sensitive or mechano-sensitive ion channel could occur for channel activation, but the energy scales in the magneto-caloric process indicate that it is theoretically feasible. It might well be that this amount of energy transfer localized to the single ferritin particle is not sufficient to locally heat up the thermo-sensitive ion channel (*Meister, 2016*; *Zheltikov, 2018*). However, it may also be that a large number of iron-loaded ferritin particles expressed throughout the neural cell membrane generate a global temperature rise (on a longer timescale) that is orders of magnitude larger than the negligible short timescale temperature rise adjacent to a single iron-loaded ferritin protein (*Zheltikov, 2018*; *Keblinski et al., 2006*).

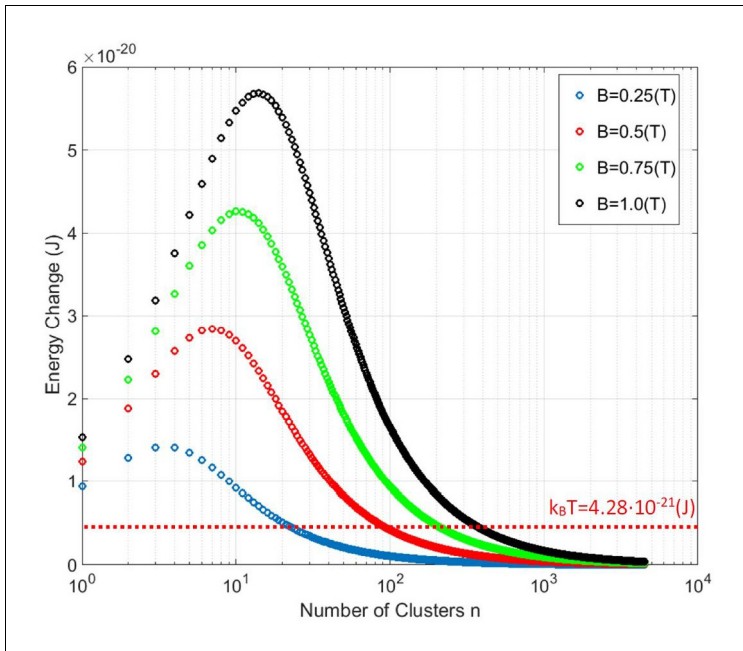

**Figure 6.** Magneto-caloric energy change for N = 4500 iron atom ferritin particle as a function of the number of clusters n (each with N/n spins) at a physiological temperature T = 310K and several applied magnetic field values B. For a given applied magnetic field B and temperature T, there is an optimal clustering size n (of N/n spins in each cluster) that will generate maximum entropy change and energy transfer to the magnetite particle lattice. No value of applied magnetic field B is sufficient to achieve magneto-caloric energy change above $k_BT$ for a paramagnetic ferritin (n = 4500, *Figure 2a*). However, grouping of the spins into n exchange coupled clusters (*Figure 2b*) achieves magneto-caloric energy changes above the $k_BT$ level.

DOI: https://doi.org/10.7554/eLife.45807.007

## Einstein-de Haas effect on iron-loaded ferritin

I now present another potential magneto-mechanical mechanism, the Einstein-de Haas effect (*Einstein, 1915*; *Galison, 1987*), that appears feasible in iron-loaded ferritin. It is a fundamental tenet of quantum mechanics that magnetic moment m of a particle is proportional to mechanical angular momentum L of that particle, m = ɣ·L, where ɣ is the gyromagnetic ratio, ɣ = e/$m_e$, for spin angular momentum of iron (where e is charge of the electron and $m_e$ is mass of the electron [*Coey, 2010*]). The Einstein-de Haas effect refers to the magneto-mechanical effect, required by the conservation of angular momentum, where a reversal of a magnetic moment of a sample by an applied magnetic field has to be accompanied by a corresponding change in mechanical angular momentum of that sample. This physical principle has been experimentally confirmed on both the macroscopic (*Scott, 1962*) and microscopic samples (*Wallis et al., 2006*), and recently also on the molecular scale (*Ganzhorn et al., 2016*). Here, I numerically evaluate the mechanical and thermal consequences of the Einstein-de Haas effect on the iron-loaded ferritin protein and therefore on the mechano-sensitive and thermo-sensitive ion channels in magnetogenetics.

*Figure 7* shows schematically the basic Einstein-de Haas principle. In the initial state, magnetic field is applied along the positive z-axis and the magnetic moment +m is aligned with the magnetic field. This moment caries mechanical angular momentum of L = +m/ɣ. As the magnetic field is reversed, the magnetic moment also reverses to -m value, which corresponds to the new mechanical angular momentum of L = -m/ɣ. This total change of angular momentum of ΔL = 2m/ɣ has to, by the law of conservation of angular momentum, be compensated for by the mechanical rotation of the magnetite particle (*Chudnovsky, 1994*) (the Einstein-de Haas effect). This change in mechanical angular momentum is proportional to the change in rotational kinetic energy (*Chudnovsky, 1994*) of the particle $\Delta E = \frac{\Delta L^2}{2I}$, where I is the rotational moment of inertia of a spherical particle of radius r, $I = \frac{2}{5} mass \cdot r^2$. Assuming the density of magnetite of 5.24x10$^3$ (kg/m$^3$), the mass of the 8nm

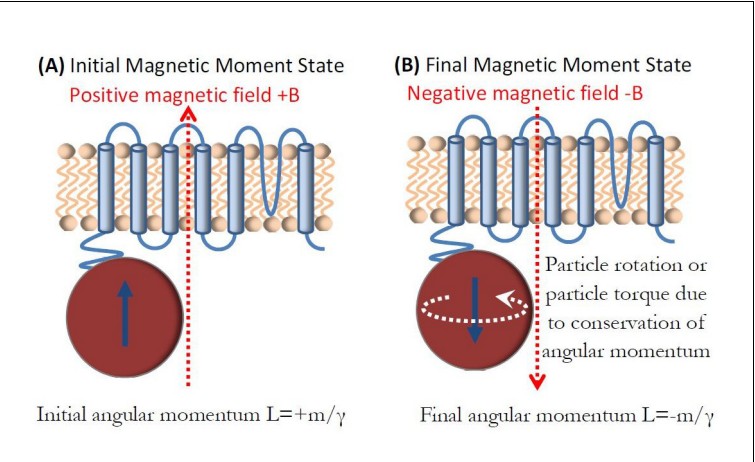

**Figure 7.** Einstein-de Haas effect in ferritin. (a) Ferritin magnetic moment +m is aligned with the field B and caries mechanical angular momentum of L = +m/ɣ. (b) Magnetic moment reversal to -m results in the total change of angular momentum of ΔL = 2m/ɣ that is compensated for by the mechanical rotation of the particle or by the mechanical torque on the particle.
DOI: https://doi.org/10.7554/eLife.45807.008

diameter particle is mass = $1.4 \times 10^{-21}$ (kg), and the energy transferred to the particle by the rotational motion imparted by the magnetic moment reversal is E = $0.33 \times 10^{-21}$ (J). This rotational kinetic energy of the magnetite particle due to the Einstein-de Haas effect has to eventually transfer to the magnetite particle lattice and the environment through friction.

Rotational kinetic energy of $0.33 \times 10^{-21}$ (J) by a single magnetite particle moment reversal (one half of the applied AC magnetic field cycle) is a significant fraction of $k_BT$ = $4.28 \times 10^{-21}$ (J). It is interesting to compare this value to the maximum energy loss per cycle of 1 (J/kg) typically reported for magnetic hyperthermia applications (*Hergt et al., 1998*; *Hergt et al., 2006*). For the Einstein-de Haas effect presented here, this value for a full single cycle is $2 \cdot \Delta E$/mass = 0.47 (J/kg). Therefore, it would appear that the Einstein-de Haas effect should be considered on par with the typical Brown and Néel relaxation modes in magnetic particle hyperthermia (*Rosensweig, 2002*; *Dutz and Hergt, 2013*; *Deatsch and Evans, 2014*) in contributing to the sample heating. As was the case for the magneto-caloric effect, it is not yet known how this Einstein-de Haas magneto-mechanical process would transfer energy to the thermo-sensitive and mechano-sensitive ion channel for activation, but the energy scales in this process are again very close to the $k_BT$ level.

This entire analysis of the Einstein-de Haas magneto-mechanical frictional heating for a magnetite particle inside the ferritin protein was predicated on the assumption that the magnetite particle is free to move inside the protein cage, a condition that to my knowledge is not presently known. So I also consider the situation in which the spherical magnetite particle is fixed inside the ferritin protein and cannot freely rotate. In this situation the change in the magnetic moment direction of the magnetite particle from +m to -m which results in the change of the mechanical angular momentum L of the particle from +m/ɣ to -m/ɣ now has to impart a torque on the surrounding medium, $\Gamma = \frac{\Delta L}{\Delta t}$, where $\Delta t$ is the time of reversal of the magnetite particle magnetic moment. A reasonable value to assume for the time of the moment reversal is $\Delta t$ = 1 (nsec), which results in the torque of 2m/($\gamma \cdot \Delta t$) = $2.4 \times 10^{-21}$ (N·m) per half cycle of the applied AC magnetic field. The energy scale of this torque is again on the same order of magnitude as $k_BT$ = $4.28 \times 10^{-21}$ (J). How this Einstein-de Haas torque would be transferred to the mechano-sensitive ion channel for activation is unknown and remains to be theoretically and experimentally explored.

## Magnetic moment fluctuations of Iron-Loaded ferritin

I finally discuss a topic of ferritin particle magnetic moment fluctuations (the topic that has generally been omitted in the discussion of magnetogenetics) and its potential effect on the ion channels. As described earlier, for the potentially superparamagnetic spin arrangement (*Figure 2c*) of the ferritin

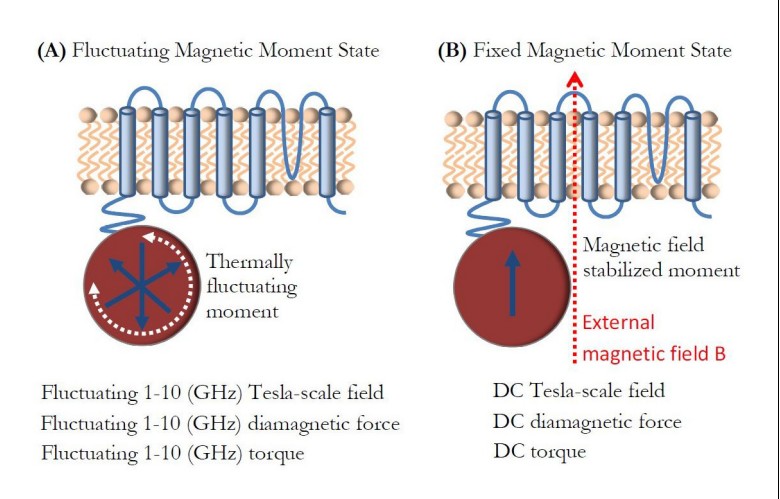

**Figure 8.** Magnetic moment fluctuations. (a) In zero external magnetic field, the ion channel experiences Tesla-scale magnetic fields and large field gradients from the fluctuating superparamagnetic particle moment at GHz-scale frequencies, as well as the corresponding AC diamagnetic forces and torques. (b) In the external field B, the ion channel experiences Tesla-scale DC magnetic fields and large field gradients from the stabilized ferritin magnetic moment, and the corresponding DC diamagnetic forces and torques.
DOI: https://doi.org/10.7554/eLife.45807.009

particle, the magnetic field near the particle surface (*Figure 4a*) has a relatively large value of 0.65 Tesla. This magnetic field from the particle is not static, but is in fact fluctuating rapidly in time (*Brown, 1963*; *Brown, 1979*), as I show schematically in *Figure 8a*. The frequency of this fluctuation at physiological temperature is significant and measured to be in the GHz frequency range through low-frequency magnetic susceptibility, Mossbauer spectroscopy, and neutron spin-echo spectros-copy (*Kilcoyne and Cywinski, 1995*; *Allen et al., 2000*; *Casalta et al., 1999*). Such superparamag-netic moment fluctuations have also been experimentally observed on a single particle scale at low temperatures (*Wernsdorfer et al., 1997*; *Piotrowski et al., 2014*; *Hevroni et al., 2015*). Therefore, the ion channel in the vicinity of the potentially superparamagnetic iron-loaded ferritin experiences large magnetic field gradients and Tesla-scale magnetic fields at GHz frequencies, as well as the cor-responding GHz frequency diamagnetic forces and torques, as discussed earlier. Upon application of the external DC magnetic field, of say 1 Tesla, the magnetic moment saturates, and the ion chan-nel experiences DC magnetic field, field gradient, diamagnetic force, and torque, as I show in *Figure 8b*. I am presently not aware of any experimental or theoretical studies that have investigated cell membrane or ion channel behavior in such extreme conditions of combined high amplitude magnetic fields (Tesla-scale) and field gradients ($10^8$–$10^9$ T/m scale) and ultra-high frequencies (1–10 GHz scale), and yet that would appear to be the environment in which the genetically engineered ion channels in magnetogenetics might be operating. Therefore, studying ion channel properties, both theoretically and experimentally, in those conditions where the timescale of ferritin magnetiza-tion dynamics and molecular physical reorientation would be warranted before final conclusions could be made about the possibilities and limitations of magnetogenetics.

## Discussion

I have presented several physical mechanisms of magneto-thermal and magneto-mechanical interac-tions in the settings relevant to the iron-loaded ferritin particle-based magnetogenetics that to my knowledge have previously not been considered. Several of these interactions have energy scales on the order of or higher than the $k_BT$ level. I emphasize that the energies associated with these interac-tions depend strongly on the iron spin clustering assumptions within the ferritin particle and are valid only for applied DC laboratory magnetic fields in the range of 0.1–2 Tesla. Some, but by no means all, reported magnetogenetics experiments to date were performed in the conditions I describe in

my analysis (many AC magnetic fields applied in reported magnetogenetics experiments were much weaker), and therefore further control and confirmation experiments with the reported magnetogenetics constructs are warranted. Many parameters that are critical to the full understanding of magnetogenetics remain unknown and include: (1) the exact iron magnetism, spin clustering configuration, spin dynamics, and magnetic anisotropy of the ferritin construct utilized so far in magnetogenetics studies, (2) the realistic mechanical and thermal nano-environment around the iron-loaded ferritin, (3) the diamagnetic (and perhaps even paramagnetic) properties of ion channels and cell membranes next to the iron-loaded ferritin, and (4) the functional properties of ion channels under the influence of simultaneously large amplitude and ultra-high-frequency magnetic fields and field gradients. Therefore, the experimental path forward clearly demands isolation of the used magnetogentics constructs and careful measurement of the above listed properties of such constructs. This would preferably be done on a single iron-loaded ferritin protein in order to avoid the confusion due to averaging of many ferritin particles. In addition to the mentioned scanning probe methods and transmission electron microscopy (TEM) and crystallography that are already being performed on a single ferritin particle level, modern techniques of SQUID (*Cleuziou et al., 2006*; *Vasyukov et al., 2013*) and cantilever force and torque magnetometry (*Stipe et al., 2001*; *Tao et al., 2016*) have achieved spectacular magnetic moment sensitivities (down to a single Bohr magneton). It would be exciting to apply them toward the characterization of single ferritin constructs reported in magnetogenetics to evaluate both their magnetic properties (as they are critical to the magnetogenetics topic in general) and the magneto-mechanical forces and torques that can be externally applied to them (as they critically relate to the magnetogenetics mechanisms I suggest as potentially plausible in particular). Importantly, these modern ultra-high sensitivity magnetometry techniques should also be utilized towards measuring the diamagnetic and paramagnetic susceptibilities of ion channels and lipid cell membranes in order to guide further experimental progress in magnetogenetics. TEM and FRET techniques (*Cost et al., 2015*) could potentially be applied to investigate the described diamagnetic membrane nanodeformations next to iron-loaded ferritin protein upon application of magnetic fields. Finally, modern nanocalorimetry (*Fon et al., 2005*) and nanothermometry (*Jaque and Vetrone, 2012*; *del Rosal et al., 2017*) techniques have also advanced to the point that studies on individual iron-loaded ferritin thermal properties and temperature changes are in principle feasible in order to evaluate some of the proposed magneto-thermal mechanisms for magnetogenetics I suggest as potentially plausible.

It is entirely possible that additional magnetic effects on ion channels in neural cell membranes, such as the phase changes and deformations in lipid bilayers due to magnetic fields (*Yamaguchi and Tanimoto, 2006*; *Maret and Dransfeld, 1977*; *Tenforde and Liburdy, 1988*; *Kurashima et al., 2002*), gradient magnetic field effects on the ion diffusion (*Kinouchi et al., 1988*) and the resting membrane potential of cells (*Zablotskii et al., 2016*), as well as the induction of calcium influx in cells by nanomagnetic forces (*Tay et al., 2016*) could further influence magnetogenetic activation of cells. As the technical advances in genetically encoded bio-mineralization of ferritin (*Matsumoto et al., 2015*; *Liu et al., 2016*) and additional experiments are carried out with both genetically (*Hutson et al., 2017*; *Liße et al., 2017*; *Mosabbir and Truong, 2018*; *Duret et al., 2019*) and synthetically prepared magnetic particles of different material compositions (*Dobson, 2008*; *Hughes et al., 2008*; *Huang et al., 2010*; *Chen et al., 2015*; *Munshi et al., 2017*) all the possible mechanisms, parameters, and conditions in magnetogenetics should be considered before the final verdict on the possibilities and limitations of this new neuro-stimulation technology is rendered.

## Acknowledgements

This work was supported by the Howard Hughes Medical Institute. I thank Tim Harris for allowing me to explore this research topic. I acknowledge discussions on this topic with Jeffrey Friedman, Sarah Stanley, Leah Kelly, Richard Smith, Tim Harris, Ben Evans, Markus Meister, Mikhail Shapiro, Hunter Davis, Joe Kirschvink, Arnd Pralle, Jon Dobson, Scott Sternson, Loren Looger, Karel Svoboda, Jacob Robinson, Ali Güler, Manoj Patel, Alan Koretsky, Doug Morris, Stephen Dodd, and David Hunt. I emphasize that my acknowledgment of the discussions with these scientists does not necessarily equal their endorsement of the ideas and results I presented here on this controversial topic.

## Additional information

### Funding

| Funder | Author |
|--------|--------|
| Howard Hughes Medical Institute | Mladen Barbic |

The funders had no role in study design, data collection and interpretation, or the decision to submit the work for publication.

### Author contributions
Mladen Barbic, Investigation

### Author ORCIDs
Mladen Barbic (iD) https://orcid.org/0000-0003-1511-1910

### Decision letter and Author response
Decision letter https://doi.org/10.7554/eLife.45807.012
Author response https://doi.org/10.7554/eLife.45807.013

## Additional files

### Supplementary files
• Transparent reporting form
DOI: https://doi.org/10.7554/eLife.45807.010

### Data availability
All results in this study are either theoretical calculations or numerical calculations.

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
