## [Decision Letter]

Thank you for submitting your article "Possible magneto-mechanical and magneto-thermal mechanisms of ion channel activation in magnetogenetics" for consideration by *eLife*. Your article has been reviewed by three peer reviewers, all experts in magnetic materials and magnetic interactions, and the evaluation has been overseen by Reviewing Editor and respondent David Kleinfeld, with Eve Marder as the Senior Editor. One outside individual involved in review of your submission has agreed to reveal their identity: Peter Littlewood (Reviewer #1).

As Reviewing Editor, I think this is an important contribution as it bears on the utility of novel perturbation techniques to probe neuronal systems. You further take issues with past analysis (Meister, 2016) for the mechanical activation via magneto-torques or, in conjunction with TRP channels, magnetic hysteresis for heat activation. In particular, the past analysis considered only the interactions of single spins with a magnetic field. All reviewers agree with this basic point, and that the present manuscript is a welcome contribution toward understanding physical limitations to neurological measurements. Yet all reviewers have provided pointed criticisms that will lead to an improved work.

Before we dive into the weeds, I would suggest that the manuscript could start with a simple statement about interacting magnetic systems. Perhaps a brief summary for your consideration could be "The total energy of a spin system has terms related to the interaction of the individual spins with the external magnetic field, plus the interaction energy of the spins among themselves. The later contribution can be quite large, as occurs in ferromagnetism. This total energy must be compared with the thermal energy and, for interacting systems, the thermal energy may be too small to appreciably dephase the spins. Thus the possibility for experimenters to exploit the interaction of magnetic nanoparticles in vivo with reasonable (~ 1 T) external fields is not unreasonable". Also, and following reviewer comments, you need to explicitly say up front that you are arguing matters of principal, as the necessary materials have yet to be discovered or synthesized in vivo.

The reviewers raise a number of critical technical issues. I request that you answer each reviewer comment and modify the manuscript accordingly. A summary of the key issues includes:

1) Further discussion of the physics of ferritin, for which there is a substantial literature, and I would also argue for a discussion of the known classes of all iron/nickel/cobalt/rare earth-containing compounds that could be seized for use in Biology.

2) Analysts of the interaction of ferritin, and other molecules, that does not make use of a quasistatic approximation. As noted by Littlewood, the relaxation time is short, i.e., estimated at less than 100 fs, and is far shorter than the period for the RF excitation in many of the experimental papers that you discuss.

3) The discussion about prior experiments is fuzzy and needs to be clearly stated. As noted by reviewer 2, you agree that many claims seem unreasonable for the interactions between ferritin and the Earth's approximately 50 mT magnetic field. This disagreement should be stated clearly. You then raise the additional point that the effects of interactions could be seen in approximately 1 T magnetic fields. The later are readily obtained in the laboratory, even easily obtained for localized, pulsed fields. Make this new claim stand out clearly.

4) A discussion – even brief – on the prospects for materials with large interactions being synthesized.

5) A discussion of an experimental path forward would be very useful and powerful way to conclude the manuscript.

*Reviewer #1:*

This paper addresses a controversial topic of magnetogenetics, which has been roundly criticized as being physically unreasonable. The main criticism is that because the moment of a ferritin molecule is so small, the magnetic susceptibility at room temperature is tiny, and there is no plausible physical effect that could arise in the typical magnetic fields that have been applied. This paper starts by accepting the basic criticism of Meister, 2016, and then systematically looks for physical conditions that would overturn the criticism. The paper is well-written, thoughtful, and I have few complaints about the analysis which has been performed. I will suggest that there is an avenue of investigation that has been missed, however.

The paper points out that the magnetic structure of Fe compounds is complicated, and that indeed one can have antiferromagnetic, ferromagnetic, and indeed more complex magnetic ordering of Fe compounds. A hypothesis is presented that in fact the ferritin core might allow for ferromagnetic configurations so that a "cluster" ferromagnet is the ground state with a much larger moment than has been traditionally assumed for ferritin (and indeed measured in ex vivo). I can't comment on the biological plausibility of such an arrangement, except to note that it will require very different atomic arrangements in the molecule, more akin to magnetite than what is generally accepted for ferritin. But I agree that if this is accepted, then by having much larger ferromagnetic clusters, it’s not too hard to overcome the kT problem. So here I see the issue as an experimental one – isolate the ferritin and measure its magnetic properties.

Indeed, once one accepts that there is a FM core, one can also generate a magneto-caloric effect (although this is generally a much larger effect near the critical point of an AFM) as well as the Einstein-de Haas effect. What I will remark in general is that all of these effects are potentially relevant provided they are carefully tuned so that the magnetic energy is comparable to the thermal energy, and that is fundamentally the game here. As a theoretical exercise, this paper is fine.

I am puzzled, however, that this paper and previous discussions as well (no less Meister, 2016) don't focus at all on what we know about ferritin, which has been rather well studied as a physical molecule (if for non-biological reasons). Conventional "horse-spleen ferritin" seems to be a strong antiferromagnet (i.e. its effectively ordered up to apparently high temperatures) with a small uncompensated moment that is much smaller than the ordered antiferromagnetic moment. The physics of ferritin doesn't have much to do with its small residual magnetic moment. I've seen no studies that suggest that ferritin has the kind of cluster ferromagnetism suggested in this paper. In fact, ferritin is so strongly ordered at low temperatures that the only mechanism to "flip" the spin of this molecule below a few tens of Kelvin is quantum mechanical tunneling. Anyway, the magnetization dynamics of this system as an ex vivo molecule have had a lot of attention in the condensed matter physics literature – for example D. D. Awschalom et al., 1992., Tejada et al., 1997, but indeed there is quite a lot of literature that's not been addressed in this paper or the previous ones.

Because there is an accidentally(?) small overall magnetic moment, in an applied field the overall magnetic configuration can flip (assuming that the molecule itself is stationary, as we will see a generally good approximation). This has been measured. Fits to a simple Arrhenius plot (Kilcoyne and Cywinski, 1995) give an activation temperature of 318K and an characteristic attempt time of 9x10^-12^ seconds. This comes from direct measurement up to a few kHz, and also Mossbauer – corresponding to probably 100 MHz. If this can be extrapolated to 300K (i.e. the system remains antiferromagnetically well ordered) then the characteristic frequency F_0_ would be on the order of 30 GHz (or the characteristic time scale 30 fsec). This is quite a long time-scale on a molecular size – for example a sound wave in water propagates about 50 nm in 30 fs.

Anyway, the point is that on a time scale shorter than 1/F_0_ the ferritin will be magnetically frozen, and in particular there will be populations of magnetization that (briefly) show hysteresis. This time scale is almost certainly shorter than the time scale for the molecule to physically reorient (by rotation, like a bar magnet) and it is not so different from the natural time scales for molecular motion on the relevant time scales for an ion channel. In consequence, I am not convinced by any analysis that assumes a quasi-static equilibrium population of "up" and "down" moments.

Since this is a very speculative topic, let me throw in something else. It's not at all implausible that the typical anti-ferromagnetic ordering temperature could be in the range of biological relevance. There are well-described physical systems where modest magnetic fields on the 1T scale modulate such phase transitions, and indeed have consequences in macroscopic properties such as giant magnetocaloric effects, and abrupt changes in conductivity and electrical screening (for example much studied in manganites). I find it interesting to note that the characteristic time scale for magnetic rearrangement in ferritin hits a microscopic scale just around room temperature, and – biology being what it is – perhaps this is not an accident.

What would be helpful on this topic would be a careful experimental materials science study of this system, rather than having too many more speculative theory papers. Coming back to the present paper, I think it’s useful, though in my view incomplete.

*Reviewer #2:*

This paper by Mladen Barbic analyzes several physical mechanisms that could enable "magnetogenetics"-the control of neuronal activity with magnetic fields. The paper is somewhat of a response to "Physical limits to magnetogenetics" in *eLife* by Markus Meister. In that paper, Meister goes through a few convincing arguments that the magnetic fields, forces, and energies associated with magnetogenetic tools are orders of magnitude smaller than those necessary to sufficiently magnetize a ferritin protein or open an ion channel. Barbic's paper verifies many of Meister's arguments, using a more detailed approach to his calculations. Barbic argues though that if one considers the possibility of ferritin being superparamagnetic, then with strong, experimentally realizable, magnetic fields, one may reach the force values necessary to open ion channels. Meister does acknowledge the possibility of superparamagnetic ferritin but doesn't take it into account in his calculations. Perhaps oddly, Barbic does not refute Meister at all – indeed the previously reported experimental magnetogenetic studies still lie almost entirely outside of the parameter regime where Barbic suggests that external magnetic fields could open a channel coupled to ferritin.

In this way, Barbic's comparison to experiment is unsatisfying. He presents scenarios that parallel the experiments in some ways (for example, a single ferritin with 4500 Fe atoms coupled to an ion channel), but then argues that a response is plausible in a magnetic field that is notably stronger than the ones used in the same experiments (e.g. 1 Telsa vs. 50 mTesla in the experimental papers). What, then, does Barbic make of the experimental data, and how valuable are his comparisons to the experiments? Considering all these things, Barbic's paper, which is quite reasonable, has a strange effect: it presents itself as suggesting that experimental magnetogenetic results are plausible, but then shows arguments that, in my view, discredit the proposed mechanisms in the experimental magnetogenetics papers. I would urge Barbic to contend more specifically with the previous experimental results.

Despite what I said above, I think this paper is a contribution to the literature on magnetogenetics. It does computations where Meister provided back-of-the-envelope numbers and explores a much wider array of magnetization for ferritin particles. Moreover, it presents new physical mechanisms that could enable magnetogenetic manipulation with the systems that have already been created. Regarding the new mechanisms proposed by Barbic, I believe he should expand his analysis of diamagnetic forces and include more details, as I outline in the notes below. The magneto-caloric effect seems a bit less convincing. Similarly, the Einstein-de Haas discussion seemed somewhat plausible, but not exciting. Perhaps it would be interesting for the author to propose experiments that would test whether or not magnetogenetic effects are due to these physical mechanisms?

*Reviewer #3:*

This paper discussed several magneto-mechanical and magneto-thermal mechanisms that the author theorized that may potentially account for the reported ion channel activation in magnetogenetics. There have been several recent reports of the ability to activate ion channels (e.g. TRPV1 and TRPV4) that are fused with ferritin using DC and AC magnetic fields. The mechanisms of these observed effects are actively being debated without a consensus being reached. Critically, some argued that the interaction between ferritin and the applied magnetic field is not sufficient to produce the needed heat or force for channel activation. In this paper, the author considered a number of hypothetical scenarios that may generate the needed heat or force. The nature of this consideration is theoretical; the discussion is timely. It provides some additional directions in searching for the underlying mechanisms for the reported effects.

1) The proposed mechanisms seem to all involve the application of a strong static magnetic field. In the cited literature Stanley et al., 2012, 2015, the vast majority of the experiments were done with RF waves at 100s KHz. About one half of the experiments in Stanley et al., 2016, were also done at RF frequencies; the rest were done at the fringe field (0.2-0.5T as reported) of a 3T MRI scanner. Wheeler et al., 2016, was done with static fields (< 1T). Other related work used different types of magnetic fields, e.g. Hutson et al., 2017 was done at RF field at μT range, while Mosabbir and Truong, 2018, used a pulsed DC field at mT range. Since the proposed mechanisms require a strong static field, it will be necessary to make a distinction between these experimental conditions.

2) The work by Mosabbir and Truong, 2018, used a static electromagnet that was programmed to be 5 s on and 2 min off on the order of 10 mT. This field seems to be much smaller than what is needed for the proposed mechanisms.

3) The proposed mechanisms all require the ferritin to be highly loaded (if not full) with iron and the iron spins to be highly coupled to produce a large magnetic moment. The direct experimental evidence of such configurations seems to be lacking. If one were to go about testing this hypothesis, what experiments would the author suggest?

4) For the mechanism evoking Einstein-de Haas effect, the magnetogenetic experiments that have been reported so far involving mostly the application of slowly changing magnetic field that does not induce rapid flipping of spins. Further, when RF fields are applied, the magnetic field at mT or μT will likely not be strong enough to flip the spins all together. The quoted energy of 0.33x20^-21^ J has to come from the RF waves which do not seem to have that kind of energy density.

5) In general, a suggestion of testable experiments would be helpful for evaluating each mechanism proposed.

---

## [Author Response]

*Before we dive into the weeds, I would suggest that the manuscript could start with a simple statement about interacting magnetic systems. Perhaps a brief summary for your consideration could be "The total energy of a spin system has terms related to the interaction of the individual spins with the external magnetic field, plus the interaction energy of the spins among themselves. The later contribution can be quite large, as occurs in ferromagnetism. This total energy must be compared with the thermal energy and, for interacting systems, the thermal energy may be too small to appreciably dephase the spins. Thus the possibility for experimenters to exploit the interaction of magnetic nanoparticles* in vivo *with reasonable (~ 1 T) external fields is not unreasonable". Also, and following reviewer comments, you need to explicitly say up front that you are arguing matters of principal, as the necessary materials have yet to be discovered or synthesized* in vivo.The reviewers raise a number of critical technical issues. I request that you answer each reviewer comment and modify the manuscript accordingly. A summary of the key issues includes:1) Further discussion of the physics of ferritin, for which there is a substantial literature, and I would also argue for a discussion of the known classes of all iron/nickel/cobalt/rare earth-containing compounds that could be seized for use in Biology.2) Analysts of the interaction of ferritin, and other molecules, that does not make use of a quasistatic approximation. As noted by Littlewood, the relaxation time is short, i.e., estimated at less than 100 fs, and is far shorter than the period for the RF excitation in many of the experimental papers that you discuss.3) The discussion about prior experiments is fuzzy and needs to be clearly stated. As noted by reviewer 2, you agree that many claims seem unreasonable for the interactions between ferritin and the Earth's approximately 50 mT magnetic field. This disagreement should be stated clearly. You then raise the additional point that the effects of interactions could be seen in approximately 1 T magnetic fields. The later are readily obtained in the laboratory, even easily obtained for localized, pulsed fields. Make this new claim stand out clearly.4) A discussion – even brief – on the prospects for materials with large interactions being synthesized.5) A discussion of an experimental path forward would be very useful and powerful way to conclude the manuscript.

All of the changes, corrections, and additions to my manuscript per Reviewing Editor and reviewers’ suggestions are clearly highlighted throughout the revised manuscript. The summary of these is listed here:

1) I included the simple statement in the introductory part of the manuscript about the interacting magnetic system, per reviewing editor’s suggestion. I now clearly state that the manuscript argues the topic as a matter of principle, also per reviewing editor’s suggestion.

2) I am in strong agreement with all the reviewers that the major challenge now in magnetogenetics is an experimental one of isolating and measuring ferritin structurally and magnetically, hopefully on a single particle level. I performed a wide literature search on the experimental methods that might be suitable for such a task, and the manuscript now includes an extensive discussion on the potential experimental path forward, per all the reviewers’ and editors’ requests. Where appropriate, I now clearly state what experimental technique might be particularly appropriate for testing some of my model predictions.

3) I now clearly state the experimental range of validity of my calculated estimates, and clearly state that some, but by no means all, previous experiments in magnetogenetics fall into that experimental range, and therefore require further careful experimental study and confirmation.

4) Upon receiving the reviewers’ request, I performed an extensive literature search on the previous physics and materials science findings on ferritin, and all of those relevant citations that the reviewers requested are now cited in the revised manuscript. Where appropriate, I now clearly state where the previous experimental findings relatively closely match my model system assumptions.

5) I now acknowledge the challenge of the magnetization dynamics of ferritin, per reviewer request. I have an extended discussion on the ferritin magnetization dynamics in the manuscript now, while acknowledging clearly that this speculative topic is heavily understudied in magnetogenetics and requires further investigation.

6) Per reviewer request, I now have a more extended discussion on the estimate of diamagnetic deformation, and I include a new more specific order of magnitude calculation based on the results from contact mechanics that elaborates on my model estimation and further justifies it. Per reviewer suggestion, I added further references that bolster that model suggestion.

7) Per reviewing editor request, I now include a brief discussion on the wider prospects of using synthesized materials with large interactions that might be possible with ferritin protein. I have added numerous citations to bolster that discussion and suggestion.

8) Where appropriate, as suggested by the reviewers, I have modified and simplified the relevant figures. I have added more labels that the reviewers pointed out were needed and removed some redundant or misleading lines and arrows in some of the figures that the reviewers pointed out were confusing.